# Sensor Integration in a Forestry Machine

**DOI:** 10.3390/s23249853

**Published:** 2023-12-15

**Authors:** Tiago Pereira, Tiago Gameiro, Carlos Viegas, Victor Santos, Nuno Ferreira

**Affiliations:** 1Engineering Institute of Coimbra (ISEC), Polytechnic of Coimbra (IPC), 3030-199 Coimbra, Portugal; tiagocostagameiro@gmail.com (T.G.); vsantos@isec.pt (V.S.); nunomig@isec.pt (N.F.); 2ADAI (Association for the Development of Industrial Aerodynamics), Department of Mechanical Engineering, University of Coimbra, 3030-788 Coimbra, Portugal; carlos.viegas@uc.pt; 3INESC Coimbra (Institute for Systems and Computers Engineering at Coimbra), 3000-033 Coimbra, Portugal; 4GECAD—Knowledge Research Group on Intelligent Engineering and Computing for Advanced Innovation and Development of the Engineering Institute of Porto (ISEP), Polytechnic Institute of Porto (IPP), 4200-465 Porto, Portugal

**Keywords:** sensors, ROS, forestry machine, sensor fusion

## Abstract

This paper presents the integration of multimodal sensor systems for an autonomous forestry machine. The utilized technology is housed in a single enclosure which consolidates a set of components responsible for executing machine control actions and comprehending its behavior in various scenarios. This sensor box, named Sentry, will subsequently be connected to a forestry machine from MDB, model LV600 PRO. The article outlines previous work in this field and then details the integration and operation of the equipment, integrated into the forest machine, providing descriptions of the adopted architecture at both the hardware and software levels. The gathered data enables the assessment of the forestry machine’s orientation and position based on the information collected by the sensors. Finally, practical experiments are presented to demonstrate the system’s behavior and to analyze the methods to be employed for autonomous navigation, thereby assessing the performance of the established architecture. The novel aspects of this work include the physical and digital integration of a multimodal sensor system on a forestry machine, its use in a real case scenario, namely, forest vegetation removal, and the strategies adopted to improve the machine localization and navigation performance on unstructured environments.

## 1. Introduction

Forest fires have had a significant and increasing impact in recent years, fueled by the effects of global warming and threatening more and more regions of the globe. They result in extensive environmental damage, including water pollution, vegetation and biodiversity loss, and a decrease in air quality [1]. A study conducted by researchers at the University of Maryland identified that forest fires lead to the loss of 3 million forest hectares annually. The year 2021 was considered one of the worst years in terms of forest fire damage, with a loss of 9.3 million hectares of forest land, worldwide [2]. Forest fires have a negative economic impact, leading to the loss of primary resources provided by the forest. Additionally, they contribute to the abandonment of rural areas, resulting in the reduced surveillance of forested zones, a decrease in jobs, and a decline in tourism [3].

Therefore, it is necessary to implement mechanisms to prevent or minimize the risk of fires. One of these mechanisms involves the management and removal of forest biomass, to create fuel or fire breaks, effectively reducing the fire risk [4]. Consequently, every year, new laws and applicable regulations are created by the government compelling institutions and individuals to clear forested areas to prevent or reduce the risk of fires [3].

Forestry operations such as vegetation removal can be accomplished through handheld biomass cutting equipment, but most efficient methods involve large forestry machines equipped with mulching apparatus [5]. These machines require one human operator per machine, who can be on the machine itself or at a small distance, controlling through a radio control. Nevertheless, operating this type of machinery is extremely hazardous and demands specialized technical skills, as well as a responsible approach to handling such large equipment that can easily lead to serious accidents. Hence, it is imperative to implement technological resources that allow operators to work safely, minimizing the risks of accidents when operating this type of machinery [6].

This work presents a hardware and software architecture, named Sentry, engineered to automate forestry machines. The Sentry is a modular box comprising multiple sensors, communication and data processing hardware, the purpose of which is to provide perception and autonomous operation capabilities to the forestry machine. A range of methods is implemented to enable the robot’s autonomous navigation, ranging from representing the machine’s orientation and position to assessing which method offers the best performance for the robot to self-locate during navigation between GPS coordinates.

The main contributions of the paper are as follows:An analysis of the benefits of using autonomous Forestry Mulcher Machines for biofuel maintenance and fire prevention.The development and integration of an ad hoc hardware and software architecture into forestry machinery, and the evaluation of the operation performance under real conditions.

This paper is organized as follows. Section 1 introduces the motivation behind this work. Section 2 presents the state of the art related to forestry operations. Section 3 details the work conducted, and in Section 4, the testing results are presented. This paper finalizes with Section 5, where the conclusion and main findings are summarized and the future work is outlined.

## 2. Related Work

Robotics for forestry applications have seen significant advancements in recent years. However, progress in this field comes with a set of complex challenges, including the following:Unstructured terrain navigation [7].Locomotion systems developing for forest navigation [8].GPS limitations analysis in dense vegetation scenarios, which can hinder robot localization and mapping [9].

Notable agro-forestry applications include robots designed for agriculture, firefighting, and vegetation management. Firefighting robots need to have a robust mechanical structure capable of withstanding high temperatures, handling gases and dust, and having significant mobility due to the rugged terrain present in forest environments [10]. Robots like the French-origin Colossus [11] have emerged to assist firefighters in urban or industrial fire scenarios. This robot is equipped with a tank of 1 T capacity and a bomb capable of delivering between 1000 L/min to 3000 L/min [10,11]. Another robot for firefighting applications is the Fire OX developed by the manufacturer Lockheed Martin, which has a tank with a capacity of 250 gallons. This robot has a wide range of visual sensors such as RGB and Infrared cameras, and its operation can be through remote control [10,12].

In agriculture, there are field robots, robots for fruits and vegetables, and robots for animal treatment [13]. An example of a field robot is the RobHortic designed for pest inspection in horticultural crops. This robot has a set of cameras for environmental detection, and a Global Navigation Satellite System (GNSS) receiver connected to an industrial computer [13,14]. As for robots for fruits and vegetables, robots like BACCHUS have been developed to replicate manual harvesting operations, operating at various levels. During autonomous navigation, the robot inspects and gathers data about the surrounding agricultural environment, performing harvesting operations using two robotic arms [15]. Finally, robots for animal treatment are used for livestock farming due to the crisis in livestock production [16]. An example of a robot associated with this field is PoultryBot, which performs egg collection activities in large-scale chicken houses. This robot can navigate autonomously, monitor poultry, and avoid obstacles [13,16].

Forestry robots aim to manage vegetation using specific and appropriate tools. An example is the Ranger robot, developed in the SEMFIRE project to reduce biomass in forests. The developed system includes various robots, including a group of small Unmanned Aerial Vehicles (UAVs) called Scouts, which sweep a designated area to identify regions of interest. This information is then transmitted to the Ranger robot, a platform responsible for carrying out the vegetation removal task [17]. The forestry robot AgRob V18 was developed for the collection of forest biomass. This robot has a set of sensors to detect fauna and flora during its navigation. Its locomotion system is powered by an internal combustion engine, and, thus, sensors like IMU are affected by vibrations transmitted by the engine. Given this problem, its navigation and localization are through 3D SLAM algorithms such as A-LOAM and LeGO-LOAM [10]. Finally, a group of researchers developed a sensor system called FGI ROAMER R2, which is placed on an all-terrain vehicle (ATV). The sensor system consists of a GNSS receiver, an IMU, and a Light Detection and Ranging (LIDAR) sensor mounted on a rigid platform for forest mapping. The combination of these three sensors allows for the obtainment of the mapping and localization of the robot during its navigation [10,18].

## 3. Development

The first section presents a piece of equipment designed for an autonomous behavior, along with all its relevant characteristics. Following this, the work carried out on the Sentry is detailed, encompassing the mechanical structure, where the construction and application of the Sentry on the forestry machine are explained. Additionally, the selection of hardware and sensors is discussed, providing the rationale for their choice. Subsequently, the assembly of the Sentry and the integration of its components are demonstrated. Finally, the incorporation of an ROS into the Sentry’s sensors is also presented.

### 3.1. Forestry Machine Description

The forestry machine used in this work was an MDB LV600 PRO (manufactured by MDB, Fossacesia, Italy), as depicted in Figure 1. It comprises a radio-controlled diesel-powered unmanned platform, with a full hydraulic drive and forestry shredder attached. This machine is ideal for carrying out landscape maintenance tasks, including vegetation removal in areas adjacent to roads, as well as forests with high-density vegetation, due to its versatility and robustness [19].

The machine features a low fuel consumption, compact dimensions, and reduced weight, facilitating easier transportation even in challenging terrains. It operates using a remote control that transmits radio frequency signals corresponding to specific movements of the machine. The operator should always remain near the machine during its operation, since no sensors are installed. See Table 1.

### 3.2. Mechanical Structure of Sentry

The developed system was conceived as a plug-and-play and modular concept, tailored to the specifications of our forestry platform but capable of being installed in other platforms. This system incorporates all the sensors required for the machine to achieve various levels of autonomy. The mechanical structure of the Sentry comprises an aluminum box, designed to house all the necessary components for the system’s operation. Figure 2 depicts the integration of all the constituent elements of the Sentry system.

The placement location of the Sentry is at the rear of the machine to ensure that the cameras have a clear field of view (FOV) throughout the working operation. This FOV and the placement of the Sentry on the machine can be seen in Figure 3; it illustrates the field of view covered by the cameras and shows the positioning of the sensors on the Sentry on the machine.

### 3.3. Hardware Architecture

The Sentry was designed to perform a set of tasks, which are divided into fundamental tasks and functional tasks. Fundamental tasks are related to how the machine’s behavior is perceived during its operation, such as knowing the machine’s position, its inclination, and recognizing objects like people and trees, among others. Functional tasks are associated with controlling the machine’s actions, such as moving the tracks, turning on the crusher, raising or lowering the tool, and more. To carry out these tasks, the Sentry needs to be equipped with a data acquisition system as well as a processing system.

In this section, the elements of the Sentry system’s hardware architecture are presented. In some cases, a brief comparative analysis is carried out regarding existing products on the market.

#### Hardware Choice

The development of the Sentry’s hardware architecture involves a distributed system, as shown in the diagram in Figure 4, where a computer is used as the brain of the architecture.
(1)**Processing System**

The Sentry comprises a computer that manages all high-level processes, referred to as the system’s brain. The right choice of hardware required a rigorous process, due to the need to find a device capable of a high computing power, low power consumption, and compact dimensions. The NVidia Jetson Xavier Nx (NVidia: Santa Clara, CA, USA) and Intel NUC 1185G7 (Intel: Santa Clara, CA, USA) were pre-selected as suitable options. Table 2 provides a comparison between these two devices.

After a comparative analysis based on the Table 2 data, it was determined that the NVidia Jetson Xavier Nx was the ideal choice to be the brain of the Sentry. This minicomputer offers the ability to run algorithms, including neural networks, simultaneously with high-resolution sensor data processing. Additionally, it has a low power consumption and compact dimensions. Although slightly more expensive than the Intel NUC 1185G7, the NVidia Jetson Xavier Nx is especially suitable for systems using Artificial Intelligence, as well as for robotic systems and equipment with high-resolution sensors [20]. The NVidia Jetson Xavier Nx will serve as the central interface for all sensor devices, including GNSS, IMU, LIDAR, and cameras, and for the microcontroller responsible for controlling the machine’s actions. It will be located inside the Sentry.
(2)**Communication System**

The Sentry is designed to enable communication between the system and a user remotely through a web application. To be able to monitor the machine’s status during its operation, it is necessary to rely on a mobile cellular network to ensure that the computer is always connected and allows for the exchange of information. Therefore, it is necessary to equip the Sentry with a cellular router to connect to the internet. The router selection must ensure support for 3 G or 4 G LTE technology, have fast upload and download speeds, possess security features, and have compact dimensions. After researching equipment available on the market, two potential routers were selected: the RUT360 developed by Teltonika (Vilnius, Lithuania), and the DWR-921 developed by D-Link (Taipei, Taiwan). Table 3 provides a comparison between these two routers.

The RUT360 was chosen based on its superior performance compared to the DWR-921. It offers high download speeds, compact dimensions, low weight, and reduced energy consumption. This router meets the requirements of industrial and commercial applications, providing advanced connectivity with technologies like 4 G LTE, Wi-Fi, and Ethernet. Furthermore, it prioritizes security, with features such as a firewall, advanced encryption, and support for VPN [21].
(3)**The Controller System**

The Sentry features two central controllers: the NVidia Jetson Xavier Nx as the main controller and the Arduino Portenta (Arduino: Turin, Italy) (connected via USB) as the auxiliary controller. The Arduino Portenta was selected due to its USB power supply, simplifying the power connections, its compact dimensions, and compatibility with the Robotic Operating System (ROS), facilitating communication with the main controller. Both controllers work together to activate the tractor’s inputs when needed.

### 3.4. Hardware Architecture

The Sentry incorporates a wide range of sensors for redundancy and operational efficiency. Three sensor modules are integrated:A module that combines GPS, IMU, and magnetometer sensors to determine the robot’s position and orientation.A module with a 3D laser sensor, which allows for the scanning of the tractor’s operating area and the obtainment of location information.A module with a vision system consisting of two RGBD cameras and two thermal cameras, providing detailed information about the operating environment.

Figure 5 illustrates the interfaces and signal flows of the current sensor architecture.

#### 3.4.1. Machine Localization System

(1)
**The GNSS, the IMU, and the MAG System**


To determine the machine’s position, the Sentry is equipped with a system that provides information from GPS, IMU, and Magnetometer. Since the Sentry is designed as a plug-and-play system contained inside a single module, incorporating encoders on the tractor’s tracks was not an option. Therefore, a robust system is required to provide high-precision orientation and positioning. Based on these requirements, the Duro Inertial system was chosen. This is a dual-frequency GNSS receiver, and despite its higher cost, it fulfills the integration needs of the Sentry.

The Swift Navigation’s Duro Inertial system (Swift Navigation: San Francisco, CA, USA) combines a standard GNSS with Real-Time Kinematics (RTK) technology to enhance the positioning accuracy. This means that by using GNSS with real-time corrections, the Duro Inertial can provide highly precise positioning. It is equipped with a Bosch BMI160 (Bosch: Gerlingen, Germany), a 6DoF IMU capable of measuring the acceleration and rotation rate. This integration results in more accurate positioning, especially in complex environments. The fusion of IMU and GNSS data provides a precise estimate of the robot’s state. Additionally, the Duro Inertial features a three-axis Bosch BMM150 magnetometer with a wide measurement range and high resolution for detecting magnetic fields with precision. This system offers a quick response and real-time accurate readings [22]. In the Sentry, this equipment is positioned at the top of the box, as illustrated in in Section 3.7. With the integration of the Duro Inertial system, it is possible to determine the position and orientation of the tractor during operation.
(2)**RTK System**

The Sentry system is equipped with a RTK positioning system, which enhances the accuracy of the GNSS that uses signals from satellites to determine the location of a device. However, the precision can be affected by atmospheric interference, resulting in an average error of 2 to 4 m. In forest environments, this resolution is insufficient. RTK is a correction method that improves the accuracy by employing a fixed base station and a unit mounted on the robot. The base station transmits real-time position information to correct the robot’s location, resulting in enhanced precision [23].

In the study conducted by Ryan Moeller et al. in [23], a comparison was made between four RTK systems (Piksi Multi Evaluation Kit, Trimble R8s, Spectra Precision SP80, and Geomax Zenith35 Pro). The Piksi Multi Evaluation Kit was found to be the solution with better performance in GPS position correction, as well as being the most economical. Therefore, this RTK system was chosen and implemented within the Sentry, the base station unit was positioned in a location that allowed it to cover the robot’s working area.

#### 3.4.2. Vision System

The autonomous navigation of a forestry machine, such as the MDB LV600 PRO, requires special attention to safety issues, such as detecting people and other obstacles. It is crucial to equip the Sentry with a vision system capable of identifying these potential hazards. This system consists of four cameras, two of which are RGBD and the other two are thermal cameras. As for the RGBD cameras, two Intel Realsense D435i were chosen, one mounted at the front and the other at the back of the Sentry. Regarding the thermal cameras, two Flir Adks (Teledyne Flir: Wilsonville, OR, USA) were selected, both installed at the front of the Sentry.

#### 3.4.3. Accurate Structural Perception System

For the accurate structural perception of the surrounding environment, the Sentry is also equipped with a 3D laser sensor. With this type of sensor, it is possible to geometrically represent the environment, thereby obtaining more detailed information about objects in the surroundings. This is highly beneficial, for example, for obstacle avoidance, planning necessary routes, equipment, and personnel safety, as well as collision prevention. In addition to these capabilities, this sensor also allows for the generation of three-dimensional maps and enables high-precision localization, which is another crucial aspect of autonomous navigation [24]. It also acts as a redundant perception system, in situations where vision-based systems tend to fail, such as poor lighting conditions.

The Velodyne VLP-16 Laser sensor (Velodyne: San Jose, CA, USA) has 16 laser channels, allowing for a 360-degree field of view, providing a more comprehensive view of the environment, high spatial resolution for detailed object detection, and enabling the segmentation and separation of images [25]. In the Sentry, this sensor is located at the top of the box, as shown in Figure 6.

### 3.5. Operating System

The integration of the implemented system was supported by the ROS by installing this framework on the main computer, the NVidia Jetson Xavier Nx.

All the packages associated with the sensors in the Sentry were installed, as well as the communication with the Arduino Portenta controller. Table 4 presents all the packages associated with the sensors and the reference to the source code.

Using the Duro Inertial package, it is possible to obtain topics related to IMU, GPS, and magnetometer data. With the Velodyne VLP-16 package, data regarding the LIDAR’s point cloud can be acquired. The Arduino package is crucial for establishing ROS topic communication between the main computer and the auxiliary controller. This means the Arduino is able to interpret ROS messages for controlling the machine’s locomotion system.

### 3.6. Orientation Acquisition

The orientation of the robot can be described using Euler angles or quaternions, which represent the rotation of the robot around each of these axes. In this topic, the procedures used from acquiring data from the GPS, IMU, and magnetometer sensors to the adopted method for representing the location and orientation of the machine will be discussed.

#### 3.6.1. Orientation Acquisition Methodology

The purpose of acquiring the orientation of the robot with a fixed reference frame is to calculate the heading, which is associated with the rotation of a body around the vertical axis (*Z*-axis), forming a vector that represents the direction of the robot. This parameter is crucial for the robot to move along a trajectory.

For this robot, the heading is obtained from the magnetometer data. The use of the magnetometer for heading estimation has been widely adopted, and when integrated with gyroscopes and accelerometers, it provides more reliable data [29].

To obtain a correctly estimated heading, a set of procedures needs to be followed, which can be observed in the block diagram presented in Figure 7.

#### 3.6.2. Magnetometer Calibration Methodology

Magnetometers measure the Earth’s magnetic field, which has a component parallel to the surface, always pointing to the magnetic north. They are essential for navigation and localization, allowing robots to position themselves in relation to magnetic reference points [29]. For moving vehicles, it is crucial to use three orthogonal axes magnetometers. This allows for a complete rotation of the magnetic field to a horizontal orientation. In the case of the Sentry, aligned with the MDB LV600, the *X*-axis is the frontal direction, *Z* is the vertical direction, and *Y* is the transverse direction, perpendicular to the plane defined by *X* and *Z*. The Euler angles (roll, pitch, and yaw) represent the rotations of the *X*, *Y*, and *Z* axes, respectively. Knowing the pitch and roll angles, it is possible to project the magnetic field vector onto the horizontal plane using the following matrix equation [29]:(1)$$\left[\begin{array}{c} M_{xH}\\ M_{yH} \end{array}\right]=\left[\begin{array}{ccc} \cos\left(\beta \right) & \sin(\beta )\sin(ɣ) & -\cos(ɣ)\sin(\beta )\\ 0 & \cos(ɣ) & \sin(ɣ) \end{array}\right]\left[\begin{array}{c} M_{x}\\ M_{y}\\ M_{z} \end{array}\right]$$

The parameters $M_{x}$, $M_{y}$, and $M_{z}$ are the data from the magnetometer in the robot’s frame, and $M_{xH}$ and $M_{yH}$ are the data resulting from the magnetometer in the robot’s frame projected onto the horizontal plane (*XH* and *YH* as shown in Figure 8).

However, data from a magnetometer can be influenced by two groups of interferences:Hard iron: This represents fixed or slowly varying magnetic fields over time, derived from ferromagnetic materials.Soft iron: This refers to the magnetic field generated within the device itself.

Therefore, calibration becomes crucial. It’s necessary to establish a model for magnetometer calibration, where parameters like bias and scale factor are applied. This can be modeled as follows [30]:(2)$${\hat{M}}_{xH}={SF}_{xH}\times M_{xH}+B_{xH}$$
(3)$${\hat{M}}_{yH}={SF}_{yH}\times M_{yH}+B_{yH}$$
(4)$${\hat{M}}_{zH}={SF}_{zH}\times M_{zH}+B_{zH}$$
where ${\hat{M}}_{xH}$, ${\hat{M}}_{yH}$, and ${\hat{M}}_{zH}$ correspond to the actual values of the magnetic field projection in the horizontal plane. ${SF}_{xH}$, ${SF}_{yH}$, and ${SF}_{zH}$ are the scale factors and measurement errors of the magnetometer, and $B_{xH}$, $B_{yH}$, and $B_{zH}$ correspond to the application of the bias on the magnetometer measurements.

Finally, to obtain the heading, simply perform the arctan2 based on the obtained values:(5)$$Heading=atan2\;\left({\hat{M}}_{yH},{\hat{M}}_{xH}\right)\times \frac{180}{\pi }\;degrees$$

Therefore, it is necessary to create an algorithm that allows for data acquisition and computation to obtain the heading. Figure 9 shows a flowchart that encompasses the entire process from data acquisition to obtaining the heading. The acquisition process involves rotating the magnetometer around all axes, and in this way, about 1000 points were taken on each axis. After this process, the scale factor and bias parameters are calculated, and Equations (2)–(4) are applied.

#### 3.6.3. Data Filtering

To enhance the reliability of the sensor data, it is essential to address the noise. The Exponentially Weighted Moving Average Filter (EMAF) was adopted due to its ability to prioritize recent information, which is crucial for agile responses to changes in the input data. The EMAF, a type of Infinite Impulse Response (IIR) filter, is effective in reducing inaccuracies caused by noise, due from sources such as interference and temperature variations.

#### 3.6.4. Tilt Compensation

The compensation of the magnetometer’s tilt is a crucial procedure to obtain reliable readings of the angular position of each axis relative to the Earth’s magnetic field. The tilt refers to the angle between the magnetometer’s frame and the vertical direction of the Earth. By using readings from an accelerometer, it is possible to obtain the roll and pitch angles, which can be used to correct the magnetometer [30]. Figure 10 depicts the relationship between the body coordinate system of the Duro Inertial with the parameters Xb, Yb, and Zb, and the corresponding *X*, *Y*, and *Z* axes of both the accelerometer and the magnetometer’s frames.

The rotations observed in Figure 10 correspond to the Euler angles roll (ɣ), pitch (β), and heading (α), where, by using rotation matrices, the roll angle around the *X*-axis, pitch angle around the *Y*-axis, and heading angle for the *Z*-axis can be defined.

Thus, the equations to compensate for the inclination on the *X* and *Y* axes are defined in Equations (6) and (7) [30]:(6)$$X_{H}=M_{x}\times \cos\left(\beta \right)+M_{z}\sin(\beta )$$
(7)$$Y_{H}=M_{x}\times \sin\left(ɣ\right)\times \cos(\beta )+M_{y}\times \cos\left(ɣ\right)-M_{z}\sin(ɣ)\times \cos(\beta )$$
where


$X_{H}$ corresponds to the final compensated magnetic field value on the *X*-axis.$Y_{H}$ corresponds to the final compensated magnetic field value on the *Y*-axis.$M_{x}$ corresponds to the magnetic field value on the *X*-axis previously calibrated.$M_{y}$ corresponds to the magnetic field value on the *Y*-axis previously calibrated.


To obtain the Euler angles, a method that reliably estimates the roll and pitch angles for application is needed, utilizing Equations (6) and (7). These angles can be estimated in two ways: through the accelerometer or the gyroscope. However, these sensors have their own errors. Accelerometers tend to accumulate errors and noise throughout the operation of the robot, especially when hard mounted to its mechanical structure [31].

The gyroscope directly measures the rate of rotation; therefore, by knowing the initial angle of the system and the sampling rate, the gyroscope information can be used to integrate the data and obtain the angle over time. However, this procedure works well for a short period but for long periods, it suffers from “gyro drift” [31].

The method used to ensure the short-term accuracy in gyroscope angles and the long-term stability in accelerometer angles is the complementary filter. It applies a Low-Pass Filter to the accelerometer measurements and a High-Pass Filter to the gyroscope measurements, resulting in extended stability for the accelerometer and greater stability for the gyroscope in short periods. The block diagram (Figure 11) illustrates the process, which involves collecting measurements from both sensors, applying filters, and combining the data to obtain the roll and pitch angles, subsequently used in Equations (6) and (7) [31].

### 3.7. Position Acquisition

There are several ways to describe the position and orientation of a robot. A common approach to obtaining the position is to use a three-dimensional coordinate system, where the axes represent the *X*, *Y*, and *Z* directions.

#### Orientation Acquisition Methodology

The representation of the robot’s position in the workspace can be achieved in various ways, depending on the system’s specifications. In this case, the position of the MDB LV600 PRO is determined using GPS and based on the direction vector provided by the magnetometer’s heading; the resulting representation of the robot’s position in a 2D Cartesian space is shown in Figure 12.

The GPS system can express positions in latitude/longitude or in the UTM system. The latitude indicates whether a point is north or south of the Equator (0° to 90°), while the longitude indicates whether it is east or west of the Prime Meridian (0° to 180°). The UTM system uses “north” and “east” coordinates in meters, making it easier to correlate with a flat map, thus meeting the requirements for representing the system’s orientation and aligning with the Position Model proposed in Figure 12. As referenced to in Section 3.4.1, the Sentry system is equipped with an RTK system, which can provide a position accuracy between 1 cm to 1.5 cm [32], which is acceptable for a robot with the dimensions presented in Table 1. However, the RTK system requires a line of sight to establish communication, and in dense forest environments, this can fail, leading to unwanted measurement errors that could affect the robot’s navigation. Therefore, tests and algorithms were developed to estimate the robot’s position without relying on RTK. The diagram in Figure 13 illustrates the steps to obtain the estimated X and Y coordinates of the robot.

For filtering the X and Y position data, two approaches were studied: one involves applying a Simple Moving Average (SMA) Filter, and the other utilizes a sensor fusion filter, specifically the Kalman Filter. The implementation process for each filter will be discussed in the following two topics. The application of these filters originated from a study conducted by S. Han et al. [33].

For the SMA Filter, data from the GPS is used. For the Kalman Filter, the input data comes from both the GPS and the IMU. The IMU data is integrated into the prediction step, where an estimate of the next state of the system, which is the position in this case, is obtained. The GPS data is integrated in the update step, where the estimated position generated by the IMU is compared with the GPS measurements, and the difference between these values is calculated. This establishes a correction state. The Kalman Filter then calculates the gain, determining a weight to obtain a more precise position estimate. The block diagram in Figure 14 illustrates the system, involving the integration of the IMU with the GPS in a Kalman Filter.

The Kalman Filter usage arises from a ROS package called “robot_pose_EKF,” with the source code available in [34]. Through this package, the input data from the GPS in UTM coordinates and from the IMU were manipulated. In this way, and observing Figure 14, the fusion of these two sensors will result in a combined position.

## 4. Conducted Experiments

The practical tests carried out in this work aim to test and validate all the algorithms and functionalities implemented in the sensors of the Sentry. Tests were performed on the sensory system, including tests of the robot’s orientation and position.

### 4.1. Orientation Acquisition

The practical experience to test the robot’s orientation considered verifying the magnetometer calibration and tilt compensation.

#### 4.1.1. Calibration Performance

As shown in the flowchart in Figure 9, magnetometer calibration was performed, collecting 1000 points on each axis of the magnetometer. Next, the bias and scale factor values were calculated. In Figure 15, it can observe the influence of sensor calibration, where the calibrated sensor data is compared to non-calibrated sensor data. Figure 15a shows the ellipsoids of the data taken from the magnetometer on each axis, displaced from each other and from the reference axis. Figure 15b, on the other hand, demonstrates the influence of the proposed calibration method, where it adjusts the ellipsoids on each axis so that they align with each other and move to the correct reference axis to obtain a real heading. This calibration process can be found in more detail in the Appendix A.

In the following example, the heading calculation was tested. This means verifying if the magnetic north and east of the sensor coincide with the correct heading value; in this case, if it varies between 0° and 90°, respectively. For this, a Pixhawk px4 was used (Pixhawn: Zürich, Switzerland), which is a device with multiple sensors including the GPS, IMU, and magnetometer, among others, providing precise information about the magnetic north and east, as well as the true north and east of the Earth. After calibrating the Inertial Duro magnetometer, it was placed in parallel with the Pixhawk px4 equipment with the same direction and orientation as shown in Figure 16.

This comparison indicated that there is an offset of 86° between the magnetic north of the Duro and the magnetic north of the Pixhawk. This difference is due to how the BMM150 magnetometer is installed inside the Inertial Duro. However, this offset can be easily corrected by simply applying an 86° rotation matrix around the *Z*-axis, so that the vertical axis aligns with the correct magnetic north and east. After applying this rotation matrix, the axis was aligned with the magnetic north and east as shown in Figure 17.

#### 4.1.2. Tilt Performance

The tilt test was performed in the field. A linear trajectory was executed using the machine’s teleoperation system. In this trajectory, the machine ascended a slope of approximately 40°. The values obtained for the tilt were observed along with its behavior. The robot started this course on a flat surface (Figure 18a), ascending an incline, as shown in Figure 18b, and concluding its trajectory on a flat surface (Figure 18c). The robot started this path facing magnetic north, meaning the heading was at 90°, and the user gave an input to the machine to move forward without changing direction. Based on this defined path, the goal is to verify the behavior of the heading when the robot moves on inclined surfaces, due to the presence of these surfaces in forest environments.

Figure 19 displays a graph depicting the variation in the magnetometer’s heading throughout this trajectory. It is noticeable that there’s minimal variation, which is attributed from error and sensory noise and track slippage due to the uneven terrain. However, it is obvious that the tilt function is operational, and the complementary filter can mitigate the machine’s vibrations to some extent, assisting in the compensation of the tilt calculations. However, it can be observed that the proposed algorithm for the tilt compensation is operational, and the combination of the IMU data with the magnetometer, as shown in Equations (6) and (7), helps maintain the correct heading values even on inclined ascending surfaces.

#### 4.1.3. Robot Position Test

The test to assess the robot’s position aimed to evaluate which filter, either the SMA Filter or the Kalman Filter, provided a better performance. Two tests were conducted, involving a straight trajectory, where the GPS coordinates were obtained and converted into Cartesian coordinates. This allowed us to measure how far the robot deviated from the defined path. The tests were labeled Test A and Test B, both covering a distance of 10 m. To have a basis for comparison, both tests calculated the maximum distance from the path and the root mean square for all the trajectory points.

Test A was carried out along a leveled wall. A structure was built to enable the manual movement of the Duro Inertial along this wall. This structure is depicted in Figure 20a, where the Duro Inertial is fixed to a metal frame equipped with wheels for smooth movement along the wall, as shown in Figure 20c.

The collection of points was initiated as outlined in Figure 21, and the results obtained from the conducted experiment are presented in Table 5.

The points collected during the trajectory are depicted in Figure 21, where a displacement between the Cartesian coordinates is performed to cover a distance of 10 m. In this figure, the blue line represents the path from start to finish, and the application of the SMA and Kalman position filters is also depicted. Graphically, the variation in the position along the defined trajectory is observed, revealing differences between the SMA Filter and the Kalman Filter, with the Kalman Filter showing a smaller distance from the line compared to the SMA Filter. Based on the results presented in Table 6, the Kalman Filter has a superior performance, showing a smaller deviation from the path line compared to the SMA Filter and a lower root mean square.

Test B was conducted in the field with the forestry robot. For this test, a navigation algorithm was selected, and a target coordinate was given to the Sentry system, so that it could autonomously navigate to this point. The robot started with an orientation to the east, meaning the heading was 0°. The Cartesian coordinates (10.0, 0.0) m were set. Figure 22 displays the orientation, the initial point of the robot (Point A), and the final point, where the robot should stop (Point B).

Hence, three trajectories were conducted, each going from Point A to Point B, to compare the performance of the filters (SMA and Kalman) and without any filtering. These trajectories were conducted separately with each filter to assess how they perform when the robot moves autonomously using a navigation algorithm. Figure 23 displays the trajectory that the robot made with each of the filters, and Table 6 presents the results obtained from this experiment.

The results reveal that the Kalman Filter presents a greater robustness in the filtering GPS data. There is a reduction in the maximum distance to the line of about 14 cm (between the Kalman Filter and unfiltered result, calculated as 36.7–22.5). The root mean square value in the Kalman Filter is also lower, indicating higher reliability. As for the SMA Filter, it similarly shows a low root mean square value, but its maximum distance to the line is higher (approximately 6 cm).

Figure 24 illustrates the impact of filtering (in this case, the Kalman Filter) compared to the unfiltered data, resulting in a better approximation of the position. The results of Test A and Test B reveal a better performance of the Kalman Filter. However, in Test B, the values were larger. This derives from the conditions under which both experiments were conducted. In Test A, the straight-line movement of the GPS (Duro Inertial) was ensured by the experimental setup, while in Test B, this was not controlled due to the terrain configuration and added noise from the machine vibration, causing unwanted deviations from the straight path.

## 5. Discussion of the Work Performed

This research introduces the development of a multimodal sensor system for an autonomous forestry machine, named Sentry. The hardware and software architectures of the Sentry were presented, along with the underlying design rationale.

The subsystems were designed with a focus on ensuring the safe execution of the MDB LV600 operation and optimizing the localization and navigation performance. The primary development focus of the Sentry system lies in its adaptation to different forest mulching machines, enabling them to operate autonomously. Regarding the architecture of the Sentry system, it was designed to create a universal, cost-effective system that can be adapted to other applications in unstructured environments, such as agriculture, surveillance and security, last mile delivery, among others.

The primary goal was to ensure an accurate estimation of the robot’s orientation and position when moving between coordinates. To achieve this, a method for acquiring the robot’s attitude using the magnetometer was proposed. For heading acquisition, the magnetometer needs to undergo a calibration, filtering, and tilt compensation process to adapt the directional vector to unstructured environments. Position acquisition is achieved from GPS data, where converting latitude and longitude coordinates to UTM coordinates allows for representation in a Cartesian space, with the *X* and *Y* axes aligned with the magnetic east and north, respectively, provided by the heading vector. In an effort to minimize the noise from the GPS-generated data, two filtering methods were proposed: the application of an SMA Filter and the application of a Kalman Filter. Perception and control algorithms were integrated into the ROS environment, where an architecture was developed around this framework.

In the experimental tests, the performance of the proposed methods was evaluated. In terms of the orientation, the influence of applying calibration to the magnetometer data was observed, and by applying tilt compensation, a directional vector representing the robot’s orientation was obtained. The proposed method for position acquisition, based on a study conducted in [33], allowed for a comparison between two filters, where the Kalman Filter demonstrated a better performance than the SMA Filter, offering more benefits in removing noise from the GPS and IMU, achieving a similar result to the mentioned study.

Thus, the obtained experimental results indicate that the proposed system consists of a set of variables, each processed in different subsystems within the Sentry system, allowing for the creation of inputs for application in navigation algorithms. In the case of the robot’s orientation, the calibration method performed with tilt compensation allows for the verification of a correct relationship between the magnetic east obtained by the magnetometer and the true east. Regarding the position, the experimental results suggest that the application of the proposed filters, specifically the Kalman Filter, enables noise reduction and an estimation of the position over time, thus benefiting the self-localization of the robot during its navigation in a given environment.

## 6. Conclusions and Future Work

The proposed solution allows for the adjustment of a set of parameters that can be applied in the autonomous navigation of forestry machines. As future work, several enhancements to the Sentry system could be considered. These include the following:The exploration of new methods/algorithms for orientation and position acquisition. Research and the implementation of new techniques for acquiring orientation and position data by fusing information from LIDAR, GPS, IMU, and magnetometer sensors. This could lead to more robust and accurate navigation.The integration of vision systems. Develop a vision system for object detection and measurement using RGBD and thermal cameras. This would add an extra layer of safety during robot navigation, especially in environments with complex obstacles.Vibration damping system. Given that forest machinery often generates significant vibrations from the internal combustion engine, integrating a vibration-damping system becomes crucial. This system would mitigate most of those vibrations, reducing noise from sensor readouts and increasing their lifespan.

These future steps would contribute to a more refined and capable autonomous navigation system for forestry machinery, ultimately enhancing the safety and performance in challenging environments.

## Figures and Tables

**Figure 1 sensors-23-09853-f001:**
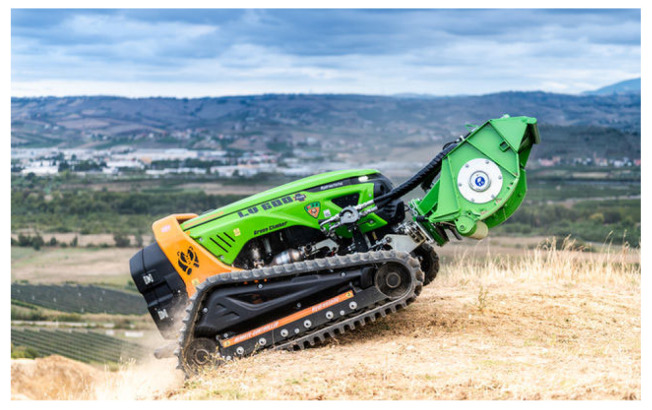
MDB LV600 PRO forestry machine.

**Figure 2 sensors-23-09853-f002:**
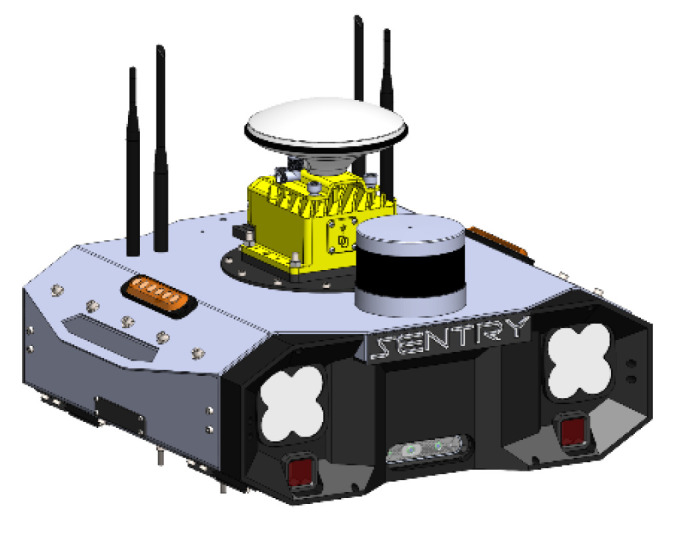
Sentry 3D model.

**Figure 3 sensors-23-09853-f003:**
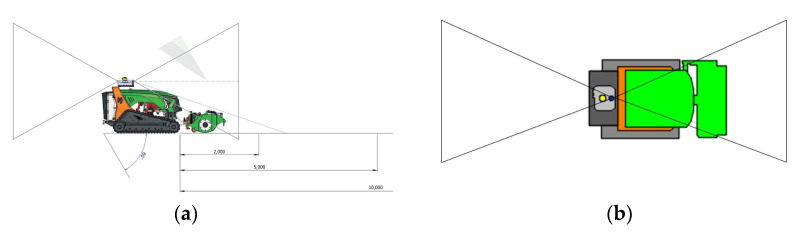
Mechanical integration of the Sentry into the LV600 PRO: representation of camera vertical FOVs (**a**,**b**).

**Figure 4 sensors-23-09853-f004:**
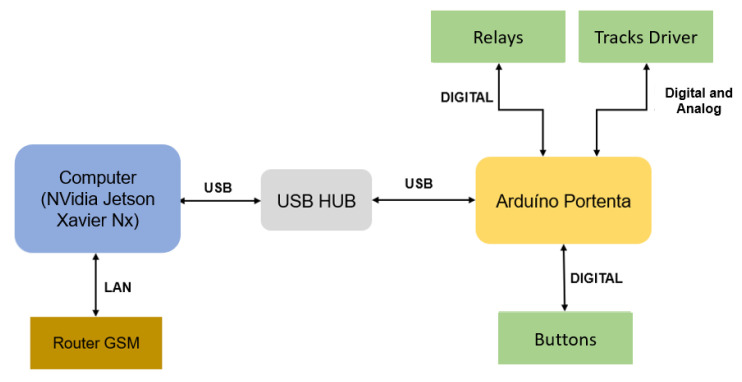
Hardware architecture of Sentry.

**Figure 5 sensors-23-09853-f005:**
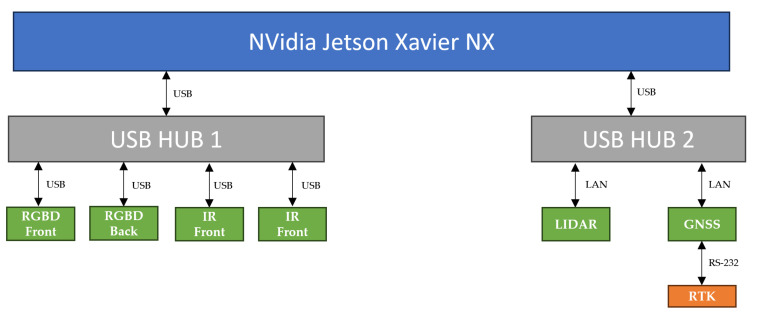
Sensor architecture of Sentry.

**Figure 6 sensors-23-09853-f006:**
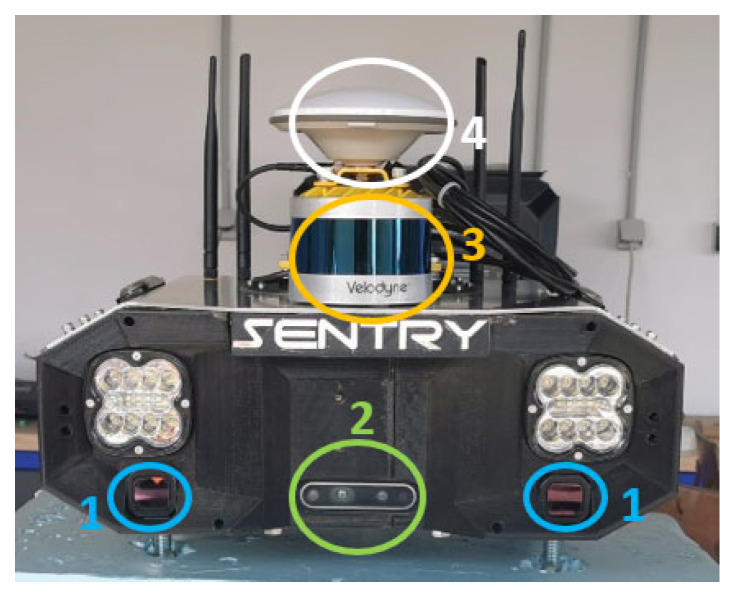
Sentry assembly, showcasing the (1) thermal cameras, (2) RGBD cameras, (3) laser sensor, (4) GNSS, IMU, and MAG modules.

**Figure 7 sensors-23-09853-f007:**
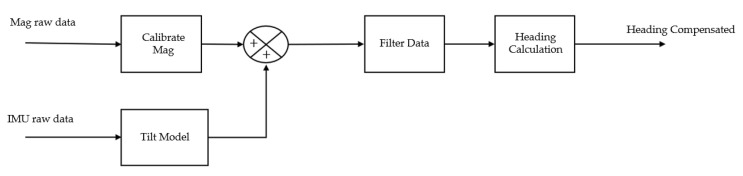
Block diagram for obtaining heading in the Sentry system.

**Figure 8 sensors-23-09853-f008:**
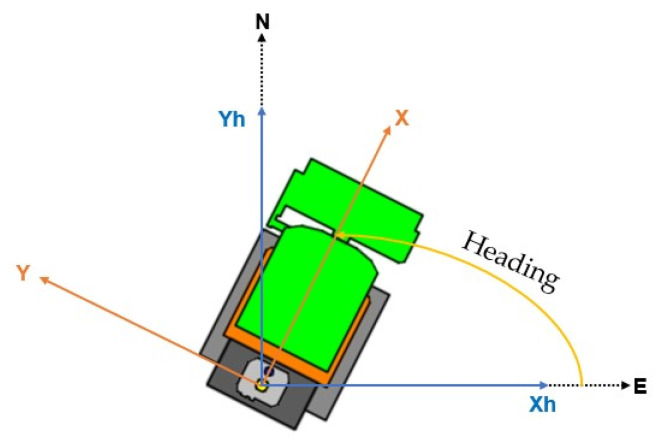
Determination of heading with a 3-axis magnetometer.

**Figure 9 sensors-23-09853-f009:**
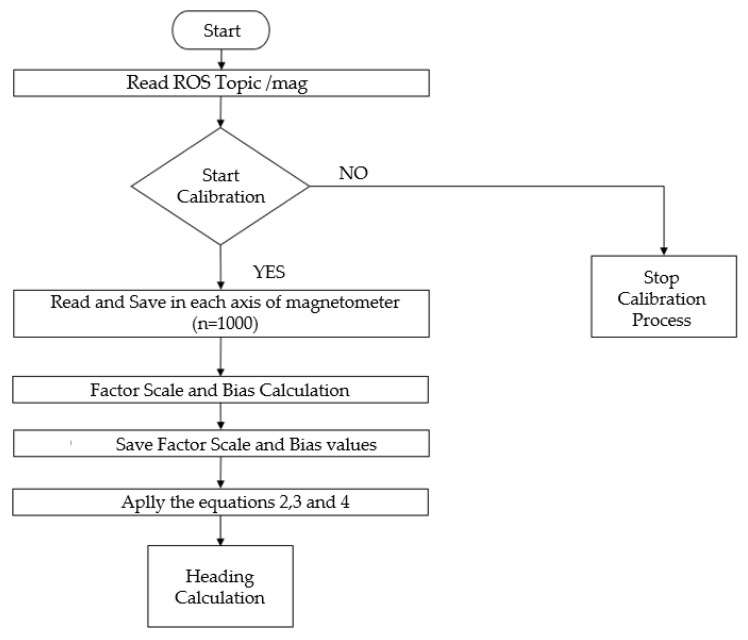
Process for magnetometer calibration.

**Figure 10 sensors-23-09853-f010:**
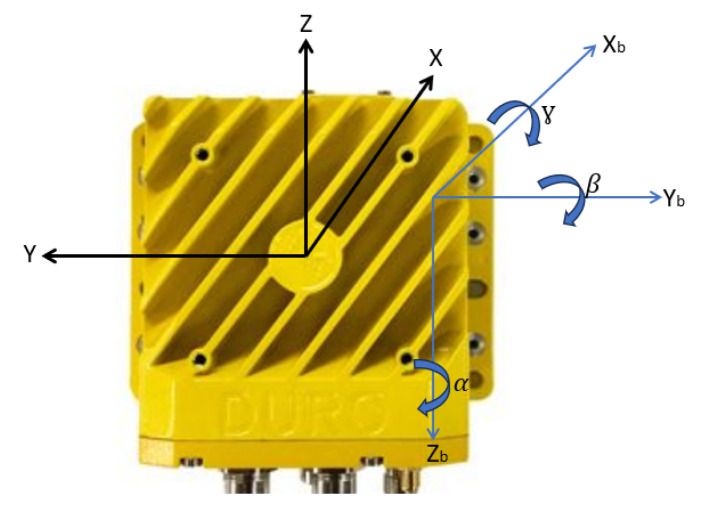
Electronic compass.

**Figure 11 sensors-23-09853-f011:**
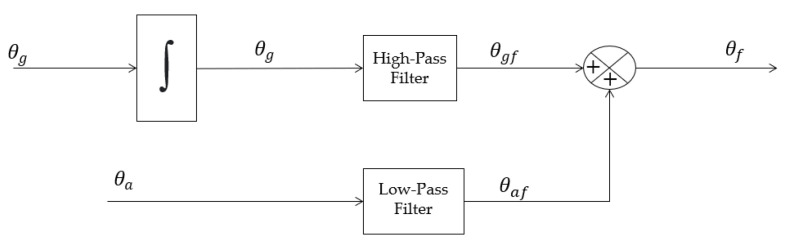
Complementary filter.

**Figure 12 sensors-23-09853-f012:**
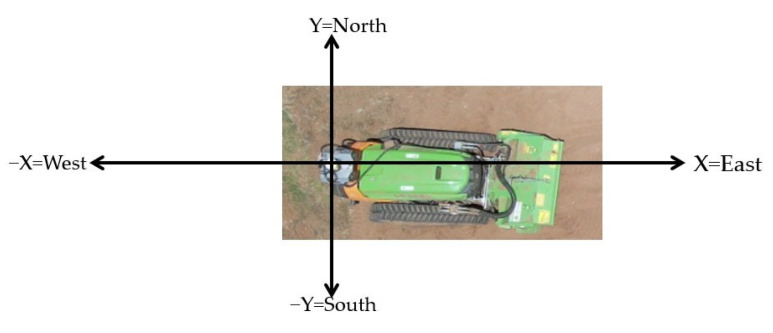
Position Model of MDB600 PRO.

**Figure 13 sensors-23-09853-f013:**
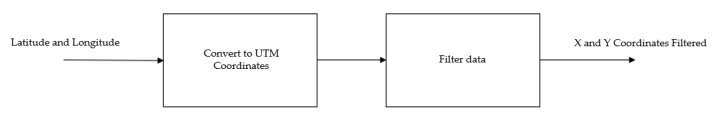
Filtered position block diagram.

**Figure 14 sensors-23-09853-f014:**
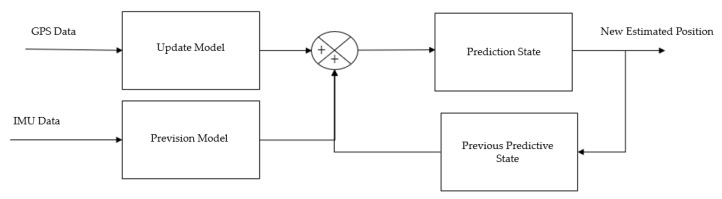
Operation of the Kalman Filter, where GPS and IMU data are fused to obtain the position.

**Figure 15 sensors-23-09853-f015:**
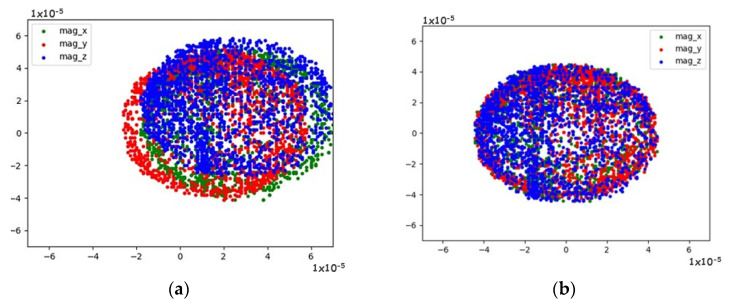
Comparison between calibrated vs. uncalibrated data: (**a**) uncalibrated data; (**b**) centered calibrated data.

**Figure 16 sensors-23-09853-f016:**
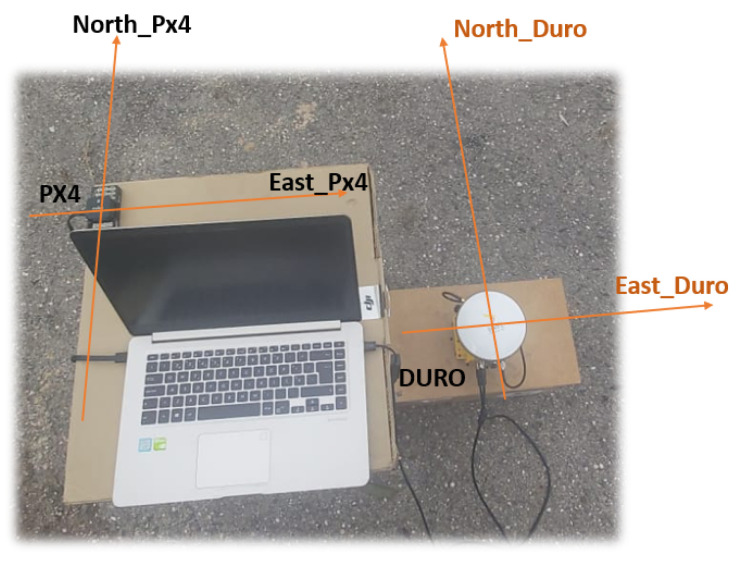
Pixhawk Px4 and Duro Inertial in parallel for heading calibration.

**Figure 17 sensors-23-09853-f017:**
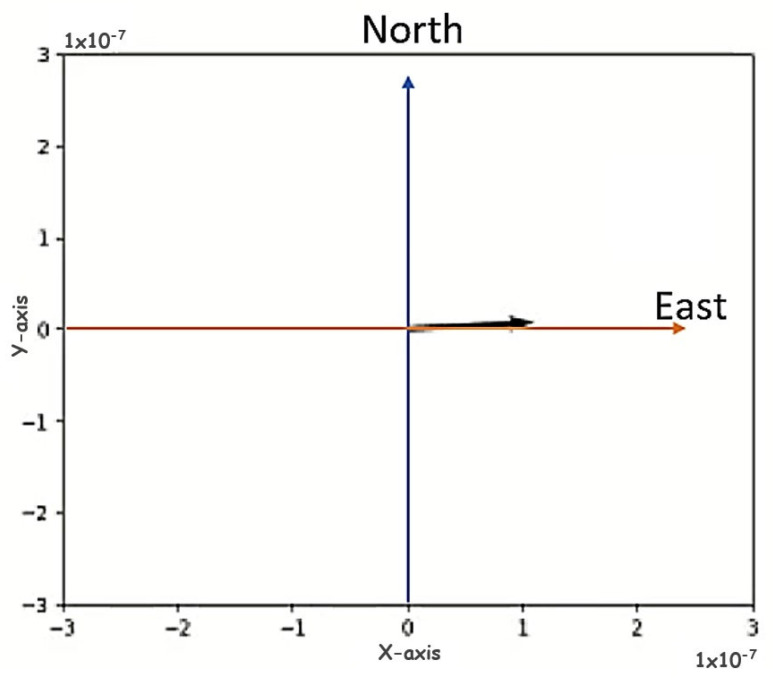
Test of magnetic east with the *X*-axis of the Duro frame. Representation of various rotations around the *Z*-axis where it was verified that magnetic north, south, east, and west coincided with the same magnetic orientation as the Pixhawk.

**Figure 18 sensors-23-09853-f018:**
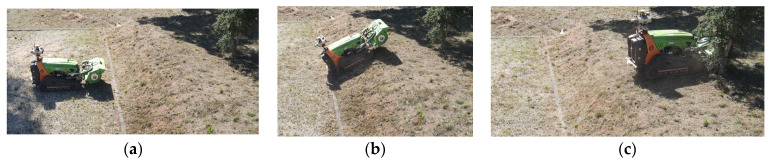
Field test for tilt verification. (**a**) initial point where the robot started. (**b**) during the trajectory, the robot was subjected to an incline. (**c**) end of the path.

**Figure 19 sensors-23-09853-f019:**
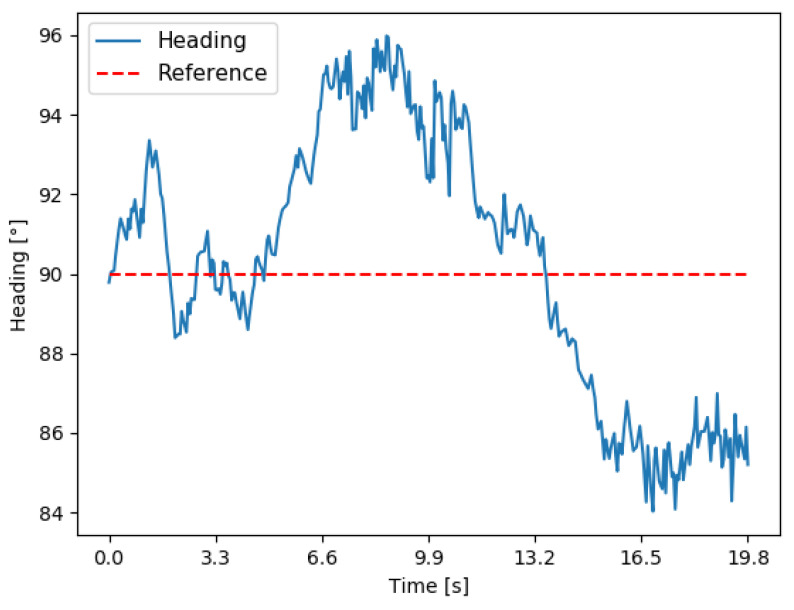
Behavior of heading during a straight-line trajectory with an inclination.

**Figure 20 sensors-23-09853-f020:**
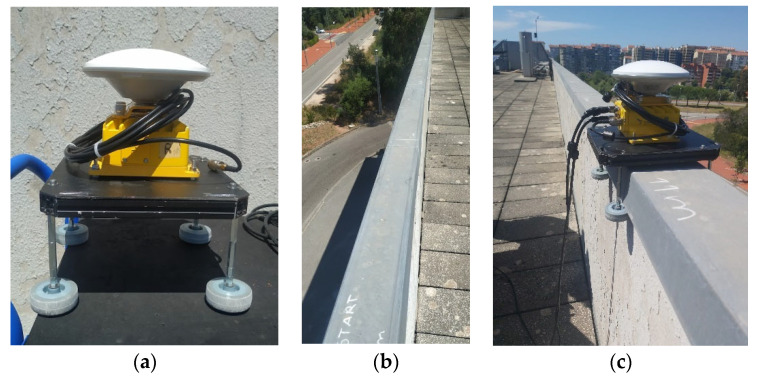
Structure built to move the Duro Inertial along a level wall (**a**–**c**).

**Figure 21 sensors-23-09853-f021:**
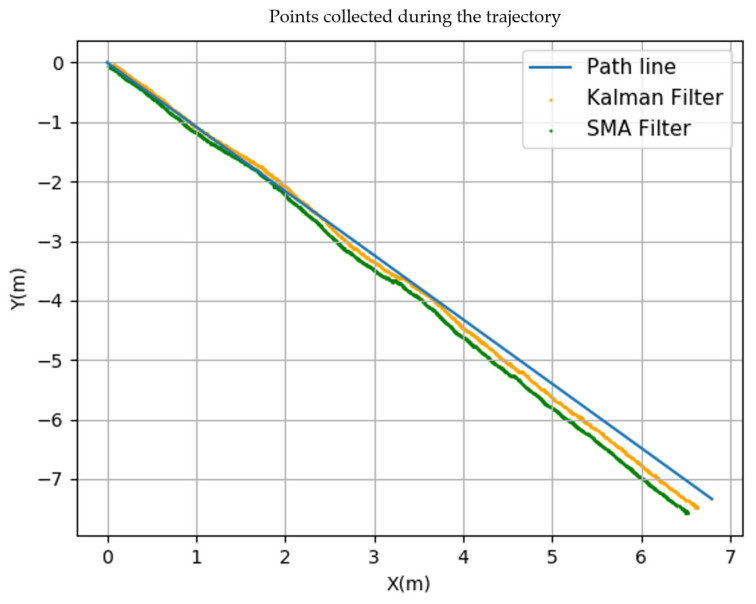
Points collected during the trajectory.

**Figure 22 sensors-23-09853-f022:**
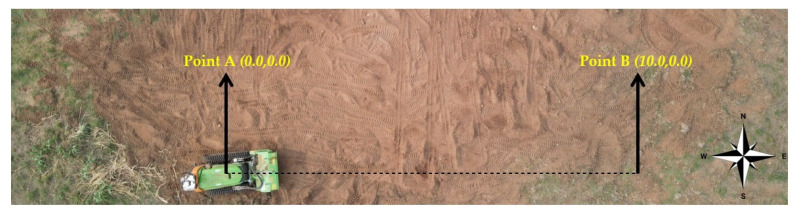
Test B conducted in the field.

**Figure 23 sensors-23-09853-f023:**
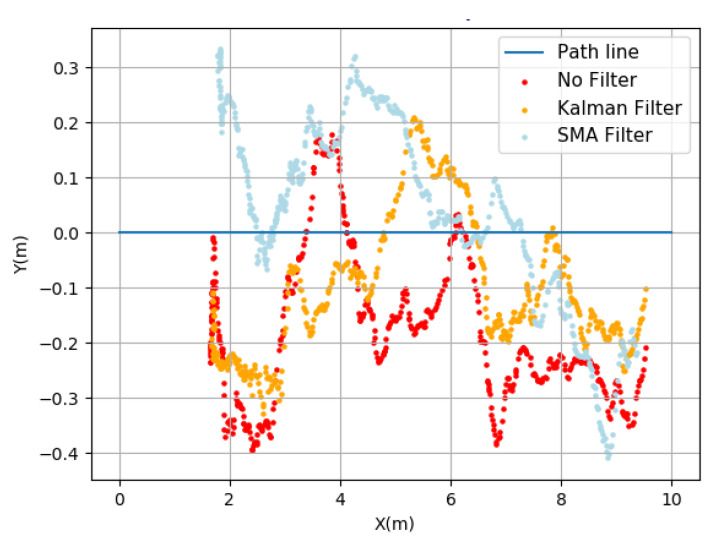
Points collected during the robot’s trajectory without filters and with the Kalman Filter and SMA Filter.

**Figure 24 sensors-23-09853-f024:**
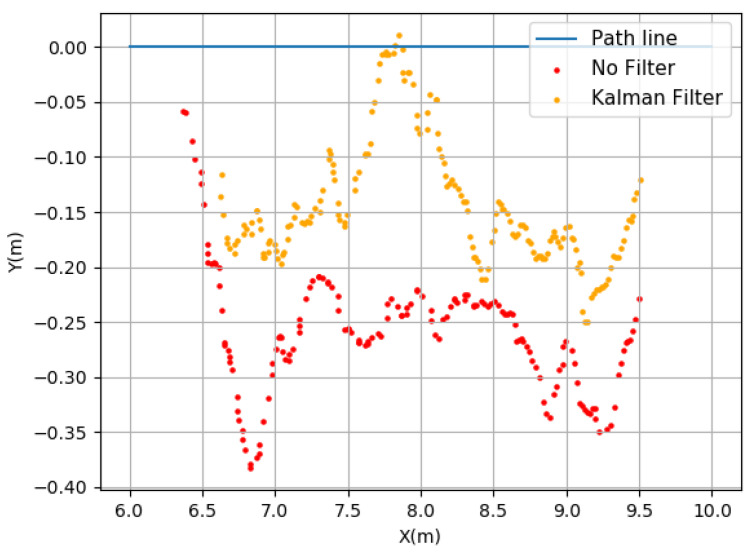
Observation of the errors present between the Kalman Filter and the absence of a filter.

**Table 1 sensors-23-09853-t001:** MDB LV600 PRO forestry machine features.

Properties	Parameters
Max inclination	60°
Transmission	Hydraulic
Speed	0–9 km/h
Width	1360 mm to 1760 mm
Length with tool	2160 mm + 750 mm
Weight with tool	1292 kg + 227 kg
Power	56 cv
Reach remote control	300 m

**Table 2 sensors-23-09853-t002:** Nvidia Jetson Xavier Nx and Intel NUC 1185G7 specifications [20].

Properties	Nvidia Jetson Xavier NX	Intel NUC 1185G7
GPU	384 NVIDIA CUDA and 48 Tensor Cores	Intel Iris Xe Graphics G7 96Eus
CPU	6-Core NVIDIA Carmel ARMv8.2 64 bit	4.8 GHz (Quad-core Intel Core i7)
Memory RAM	16 GB LPDDR4x	16 GB DDR4
SSD	256 GB	512 GB
Power consumption	15 W	28 W
Dimensions	100 mm × 87 mm × 16 mm	117 mm × 112 mm × 51 mm

**Table 3 sensors-23-09853-t003:** Comparison data between RUT360 and DWR-921.

Features	RUT360	DWR-921
Mobile technology	4 G LTE	4 G LTE
Download speed	Up to 300 Mbps	Up to 150 Mbps
Upload speed	Up to 50 Mbps	Up to 50 Mbps
Number of LAN ports	2	4
Number of WLAN ports	1	1
Wireless standard	IEEE 802.11 b/g/n	IEEE 802.11 n
SIM card slots	1	1
Security features	Firewall, VPN, WPA2-PSK	Firewall, VPN, WPA2-PSK
Dimensions (W × H × D)	100 mm × 30 mm × 85 mm	190 mm × 116 × 22.4 mm
Weight	200 g	270 g
Power consumption	10.5 W (max)	12 W (max)

**Table 4 sensors-23-09853-t004:** ROS package to communicate with sensors/components.

Sensor/Component	Package Name	Source Code
Duro Inertial	duro_gps_driver	Source code available at [26]
Velodyne VLP-16	velodyne	Source code available at [27]
Arduino Portenta	Rosserial	Source code available at [28]

**Table 5 sensors-23-09853-t005:** Maximum and RMS values obtained during Test B trajectory.

Max (cm)			RMS (cm)		
${KF}^{\left[a\right]}$	${SMAF\;}^{\left[b\right]}$	${NF\;}^{\left[c\right]}$	${KF\;}^{\left[a\right]}$	${SMAF\;}^{\left[b\right]}$	${NF}^{\;[c]}$
22.5	36.7	36.7	16.8	17.7	22.6

^[a]^: Kalman Filter, ^[b]^: Simple Moving Average Filter, ^[c]^: without Filter.

**Table 6 sensors-23-09853-t006:** Maximum and RMS values obtained during Test A trajectory.

Max (cm)		RMS (cm)	
${KF}^{\left[a\right]}$	${SMAF}^{\left[b\right]}$	${KF}^{\left[a\right]}$	${SMAF}^{\left[b\right]}$
22.05	36.7	11.0	21.0

^[a]^: Kalman Filter, ^[b]^: Simple Moving Average Filter.

## Data Availability

Data are contained within the article.

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
