# Peer review of "Sensor Integration in a Forestry Machine"

_sensors, 2023, doi:10.3390/s23249853_

Round 1

Reviewer 1 Report

Comments and Suggestions for Authors

This paper describes the development of a new sensor system for autonomous forestry machines. The system is housed in a single enclosure and contains a set of sensors that are used to control the machine and to understand its behavior in various scenarios. Finally, the paper presents practical experiments that demonstrate the system's behavior and analyzes the methods that will be used for autonomous navigation.

Graphically, the article is processed well. Scientific style is appropriate. I have the following comments on the manuscript:
I miss the state of the art chapter where the authors describe the existing sensory systems. It is necessary to better describe the contribution and novelty of your solution. In the discussion chapter, it is necessary to critically evaluate your sensor system with existing ones. A minimum of work with literature is also presented by a few scientific articles in references.
After incorporating the comments, I recommend publishing the article.

Author Response

Dear Reviewer,

Thank you for your prompt review of our manuscript titled " Sensor Integration in a Forestry Machine". We appreciate the time and effort devoted to evaluating our work. We have carefully considered the comments and suggestions provided and made the necessary revisions to address each of them.

Regarding the suggested changes, and starting with the state of the art, an enhancement was made to describe other existing technologies and sensory architectures for autonomous robots. These changes are in Chapter 2, from lines 113 to 123 with the following text "The forestry robot AgRob V18 was developed for the collection of forest biomass. This robot has a set of sensors to detect fauna and flora during its navigation. Its locomotion system is powered by an internal combustion engine, and thus sensors like IMU are affected by vibrations transmitted by the engine. Given this problem, its navigation and localization are through 3D SLAM algorithms such as A-LOAM and LeGO-LOAM [10]. Finally, a group of researchers developed a sensor system called FGI ROAMER R2, which is placed on an all-terrain vehicle (ATV). The sensor system consists of a GNSS receiver, an IMU, and a Light Detection and Ranging (LIDAR) mounted on a rigid platform for forest mapping. The combination of these three sensors allows obtaining the mapping and localization of the robot during its navigation [10, 18]."

Finally, as suggested by the reviewer, the contribution and novelty of the Sentry System were described. These changes are in Chapter 5, from lines 673 to 675 with the following text "The primary development focus of the Sentry system lies in its adaptation to different forest mulching machines, enabling them to operate autonomously".

The changes are highlighted in the document in Turquoise Blue color.

In general, we believe that the revisions we have made have significantly improved the quality and clarity of our manuscript. We hope that these changes and responses have addressed the concerns raised.

Thank you for your consideration of our revised manuscript, and we look forward to hearing back from you soon.

Sincerely,

Tiago Pereira

Tiago Gameiro

Carlos Viegas

Victor Santos

Nuno Ferreira

Reviewer 2 Report

Comments and Suggestions for Authors

1. The second chapter introduces the current agricultural application of robotics technology, but the overall literature is insufficient, and there is a lack of summary of the current situation and shortcomings.

2. What are the innovative points and main advantages of sensor integration in this article? Do you have special requirements or universality for application scenarios?

3. There is a lack of necessary explanations and explanations for the experimental images, such as Figure 15, Figure 19, and Figure 21. Please supplement and improve.

4. Lack of overall evaluation and analysis of experimental results. What are the specific evaluation indicators?

5. The format of images and tables should be consistent.

Comments on the Quality of English Language

Excellent

Author Response

We appreciate the time and effort devoted to evaluating our work. We have carefully considered the comments and suggestions provided and made the necessary revisions to address each of them.

In response to the suggested changes, and addressing each of the questions:

Question 1:
Enhancements have been made to the state-of-the-art section related to agriculture, where various types of robots and their objectives were discussed. These changes can be found in Chapter 2, from lines 95 to 106 with the following text " In agriculture, there are field robots, robots for fruits and vegetables, and robots for animal treatment [13]. An example of a field robot is the RobHortic designed for pest inspection in horticultural crops. This robot has a set of cameras for environmental detection, and a Global Navigation Satellite System (GNSS) receiver connected to an industrial computer [13, 14]. As for robots for fruits and vegetables, robots like BACCHUS have been developed to replicate manual harvesting operations, operating at various levels. During autonomous navigation, the robot inspects and gathers data about the surrounding agricultural environment, performing harvesting operations using two robotic arms [15]. Finally, robots for animal treatment are used for livestock farming due to the crisis in livestock production [180]. An example of a robot associated with this field is PoultryBot, which performs egg collection activities in large-scale chicken houses. This robot can navigate autonomously, monitor poultry, and avoid obstacles [13,16].".

Question 2:
The advantages and innovative features of Sentry were addressed to explicitly highlight the advantage of the adopted architecture. These changes are in Chapter 5, from lines 675 to 678 with the following text "Regarding the architecture of the Sentry system, it was designed to create a universal, cost-effective system that can be adapted to other applications in unstructured environments, such as agriculture, surveillance and security, last mile delivery, among others.".

Question 3:
A better explanation of Figures 15, 19, and 21 was provided. For Figure 15, more emphasis was given to the influence of the calibration process, with changes in Chapter 4, from lines 509 to 514 with the following text "Figure 15(a), shows the ellipsoids of the data taken from the magnetometer on each axis, displaced from each other and from the reference axis. Figure 15(b), on the other hand, demonstrates the influence of the proposed calibration method, where it adjusts the ellipsoids on each axis so that they align with each other and move to the correct reference axis to obtain a real Heading.".                                                    Regarding Figure 19, a more detailed explanation of the tilt evaluation process and the obtained data was provided. These changes are in Chapter 4, from lines 547 to 550 with the following text "Based on this defined path, the goal is to verify the behavior of the Heading when the robot moves on inclined surfaces, due to the presence of these surfaces in forest environments. " and from lines 560 to 565 with the following text "However, it can be observed that the proposed algorithm for Tilt compensation is operational, and the combination of IMU data with the Magnetometer, as shown in equations 6 and 7, helps maintain the correct Heading values even on inclined ascends surfaces.".  Finally, Figure 21 illustrated the influence of the Kalman Filter and the SMA Filter. These changes are in Chapter 4, from lines 609 to 615 with the following text "The points collected during the trajectory are depicted in Figure 21, where a displacement between Cartesian coordinates is performed to cover a distance of 10 meters. In this figure, the blue line represents the path from start to finish, and the application of the position filters SMA and Kalman is also depicted. Graphically, the variation of the position along the defined trajectory is observed, revealing differences between the SMA Filter and the Kalman Filter, with the Kalman Filter showing a smaller distance from the line compared to the SMA Filter.".

Question 4:
A comprehensive evaluation and analysis of the experimental results were conducted. These changes are in Chapter 5, from lines 698 to 706 with the following text " Thus, the obtained experimental results indicate that the proposed system consists of a set of variables, each processed in different subsystems within the Sentry system, allowing the creation of inputs for application in navigation algorithms. In the case of the robot's orientation, the calibration method performed with tilt compensation allows verifying a correct relationship between the Magnetic East obtained by the Magnetometer and the True East. Regarding the position, the experimental results suggest that the application of the proposed filters, specifically the Kalman Filter, enables noise reduction and an estimation of the position over time, thus benefiting the self-localization of the robot during its navigation in a given environment.".

Question 5:
Formatting improvements were also made to all images and tables in the manuscript that were not in compliance.

The changes are highlighted in the document in Green color.

In general, we believe that the revisions we have made have significantly improved the quality and clarity of our manuscript. We hope that these changes and responses have addressed the concerns raised.

Sincerely,

Tiago Pereira

Tiago Gameiro

Carlos Viegas

Victor Santos

Nuno Ferreira

Round 2

Reviewer 1 Report

Comments and Suggestions for Authors

Thanks for incorporating my comments. In my opinion, the article is excellently prepared and I recommend it for publication.